# Atomic engineering of high-density isolated Co atoms on graphene with proximal-atom controlled reaction selectivity

Huan Yan [1,2], Xiaoxu Zhao [2,3], Na Guo[4], Zhiyang Lyu[2], Yonghua Du [5], Shibo Xi[5], Rui Guo[2,6], Cheng Chen[2,6], Zhongxin Chen[2,3], Wei Liu[2], Chuanhao Yao[1,2], Jing Li [1,2], Stephen J. Pennycook[3,6,7], Wei Chen[2,4,6], Chenliang Su [1], Chun Zhang [2,4,6] & Jiong Lu[1,2,6]

Controllable synthesis of single atom catalysts (SACs) with high loading remains challenging due to the aggregation tendency of metal atoms as the surface coverage increases. Here we report the synthesis of graphene supported cobalt SACs ($Co_1/G$) with a tuneable high loading by atomic layer deposition. Ozone treatment of the graphene support not only eliminates the undesirable ligands of the pre-deposited metal precursors, but also regenerates active sites for the precise tuning of the density of Co atoms. The $Co_1/G$ SACs also demonstrate exceptional activity and high selectivity for the hydrogenation of nitroarenes to produce azoxy aromatic compounds, attributable to the formation of a coordinatively unsaturated and positively charged catalytically active center (Co–O–C) arising from the proximal-atom induced partial depletion of the $3d$ Co orbitals. Our findings pave the way for the precise engineering of the metal loading in a variety of SACs for superior catalytic activities.

---

[1] SZU-NUS Collaborative Center and International Collaborative Laboratory of 2D Materials for Optoelectronic Science & Technology, College of Optoelectronic Engineering, Shenzhen University, 518060 Shenzhen, China. [2] Department of Chemistry, National University of Singapore, 3 Science Drive 3, Singapore 117543, Singapore. [3] NUS Graduate School for Integrative Sciences and Engineering, National University of Singapore, 28 Medical Drive, Singapore 117456, Singapore. [4] Department of Physics, National University of Singapore, 2 Science Drive 3, Singapore 117542, Singapore. [5] Institute of Chemical and Engineering Sciences, 1 Pesek Road, Jurong Island, Singapore 627833, Singapore. [6] Centre for Advanced 2D Materials and Graphene Research Centre, National University of, Singapore 117546, Singapore. [7] Department of Materials Science & Engineering, National University of Singapore, 9 Engineering Drive 1, Singapore 117575, Singapore. These authors contributed equally: Huan Yan, Xiaoxu Zhao, Na Guo. Correspondence and requests for materials should be addressed to C.S. (email: chmsuc@szu.edu.cn) or to C.Z. (email: phyzc@nus.edu.sg) or to J.L. (email: chmluj@nus.edu.sg)

Single-atom catalysts (SACs) have emerged as a new frontier in the field of heterogeneous catalysis due to their remarkable catalytic performances and maximized atom utilization[1–21]. For SACs to be practically applicable, a sufficiently high loading of atomically-dispersed atoms on an appropriate support is required. Unfortunately, isolated metal atoms are thermodynamically unstable due to their high surface energy and thus prone to agglomeration at an increased loading during the synthetic process or the subsequent treatment. Common strategies to tackle this issue include reducing the metal loading to an extremely low level[1,3,9,22] and enhancing metal-support interactions for a strong anchoring of the isolated metal atoms[13,23–25]. In respect of the former strategy, the loading of the majority of SACs synthesized by wet-chemistry methods has been kept below 1% to prevent the formation of metal nanoparticles[1,4,11,22,26,27]. For instance, aggregation of Pt atoms dispersed on α-MoC surface and mesoporous zeolite substrates occurred once the loading of precious metal catalyst was increased to 0.2 and 0.5%, respectively[11,26]. In the latter strategy which involves a judicious choice of supports, the loading of Pd single atoms can be increased up to 1.5% on modified $TiO_2$ nanosheets[28]. In addition, a high loading of SACs has been achieved using a wet-impregnation[25,29] or high temperature pyrolysis method[30]. In both methods however, it was challenging to control the loading of metal atoms on the support surface for optimizing the catalytic performance. Despite considerable progress in recent years, controllable synthesis of stable SACs with sufficiently high loading for high performance catalysis remains a major roadblock towards its practical applications.

On the one hand, a strong interaction between anchored metal atoms and neighboring support atoms is essential for achieving stable high-metal-loading SACs[25,28]. On the other hand, the metal-support interaction results in modification of the electronic properties of anchored metal atoms, which in turn alters the activity and selectivity of SACs[25,31]. It is therefore of importance to probe the local coordination environment of single metal atoms and their electronic coupling with support atoms in close proximity. Such a study offers a unique opportunity to further optimize the catalytic performance of SACs, which however, remains largely unexplored.

To this end, we have devised a reliable method via the atomic layer deposition (ALD) technique for the preparation of stable high loading $Co_1/G$ SACs, which also allows the precise tuning of the density of isolated Co atoms on the graphene support. In contrast to solution-phase deposition, self-limiting surface reactions (Fig. 1, Supplementary Figure 1) of ALD ensure that each Co precursor molecule is anchored on a single active site of the graphene support[5,12,15,17]. Interestingly, the active sites on graphene can be re-generated by ozone treatment in the second pulse of each ALD cycle, allowing for the loading of another batch of Co single atoms. As a result, the loading of $Co_1$ single atoms can be precisely tuned by controlling the number of Co ALD cycles as illustrated in the Fig. 1. In the selective hydrogenation of nitrobenzene, all the $Co_1/G$ SACs prepared show outstanding activity and remarkable selectivity to azoxy compounds. The mechanistic studies show that the electronic coupling of Co atoms with adjacent oxygen atoms results in more positively charged $Co_1$ catalytic center, which helps to reduce its binding strength to azoxy compounds. Such an electronic coupling between the Co atom and its neighboring oxygen atoms prevents the full hydrogenation of nitroarenes, leading to a remarkably high selectivity towards the partially hydrogenated product.

## Results and discussion

**Synthesis of $Co_1/G$ SACs**. In our study, reduced graphene oxide was selected as the support for the preparation of $Co_1/G$ SACs

due to the following figures of merit: (i) chemically derived graphene offers an ideal low-cost platform for the anchoring of individual Co ALD precursors to the oxygen-decorated carbon sites;[32] (ii) the density of anchoring sites on graphene can be tuned by controlling the pretreatment conditions[32–34]. Under typical oxidation conditions, the basal plane of graphene can be decorated with diverse oxygen functional groups including hydroxyl, epoxy, phenolic, carbonyl, and carboxyl groups[32]. However, only specific oxygen-containing functional groups on the graphene surface are expected to act as nucleation sites to react with the metal precursors used in the ALD. Hence, it is desirable to achieve homogeneous oxidation of graphene to create a high density of identical anchor sites in order to optimize the loading density of $Co_1$ single atoms. The exposure of graphene to ozone ($O_3$) at an elevated temperature is most likely to produce uniform epoxy functional groups[34,35], which are anticipated to be active anchor sites for $Co(C_5H_5)_2$ precursors ($CoCp_2$). Furthermore, the remaining ligands of the deposited metal precursors are often removed through a combustion reaction using $O_3$ in the second pulse of each ALD cycle[36]. Hence, we expect the ozonation of a graphene support at elevated temperature to achieve two outcomes, namely allowing us to burn off organic ligands and to recreate desirable anchor sites for the subsequent Co ALD cycles, which would offer an effective method for tuning the metal loading of SACs.

To test the above hypothesis, we performed X-ray photoemission spectroscopy (XPS) measurements to investigate the evolution of the amount of oxygen-containing groups on graphene exposed to $O_3$ at 150 °C (Supplementary Figure 2). It was found that ozonation of graphene at 150 °C creates predominantly epoxy groups as confirmed by the observation of a strong O1s peak at 532.08 eV, consistent with a previous report[33]. The increase in the amount of epoxy groups on the graphene support is approximately linear in the first five cycles of ozone pretreatment but tends to plateau as the number of ozone pretreatment cycles further increases (Fig. 2g). After gaining a better understanding of the ozonation of graphene, we carried out the first cycle of Co ALD on thermally reduced graphene oxide by exposing the support to $CoCp_2$ vapor as illustrated in Fig. 1. Subsequently, molecular $O_3$ was injected into the chamber to remove the ligand and to simultaneously recreate new active sites for the loading of another batch of Co atoms in the subsequent cycle of ALD. By repeating this stepwise deposition, the loading density of $Co_1/G$ catalysts can be precisely tuned by controlling the number of Co ALD cycles. Through this method, we managed to synthesize a series of $Co_1/G$ catalysts with Co loadings of 0.4, 0.8, 1.3, 2.0, and 2.5 wt% (designated as $Co_1/G$-0.4, $Co_1/G$-0.8, $Co_1/G$-1.3, $Co_1/G$-2.0, and $Co_1/G$-2.5) by performing 1, 2, 3, 4, 5 cycles of Co ALD respectively.

**Characterization of $Co_1/G$ SACs**. State-of-the-art aberration-corrected scanning transmission electron microscopy – annular dark field (STEM-ADF) measurements were initially conducted to gain a detailed understanding of the morphologies of the as-prepared $Co_1/G$ SACs. The employed acceleration voltage is 60 kV, where the lower voltage significantly reduces the cross section of Co atoms dissociation or clustering events. The large-field of view STEM-ADF images of the as-prepared $Co_1/G$ SACs revealed the absence of larger clusters for all the samples prepared within the five cycles of Co ALD as shown in Fig. 1f and Supplementary Figure 3. Compared to the bare graphene (Supplementary Figure 4), atomic resolution STEM-ADF images revealed that Co atoms in $Co_1/G$-0.4 (Fig. 2a) and $Co_1/G$-0.8 (Fig. 2b) prepared in the first and second cycle of Co ALD respectively are atomically dispersed and well-separated on graphene without aggregation

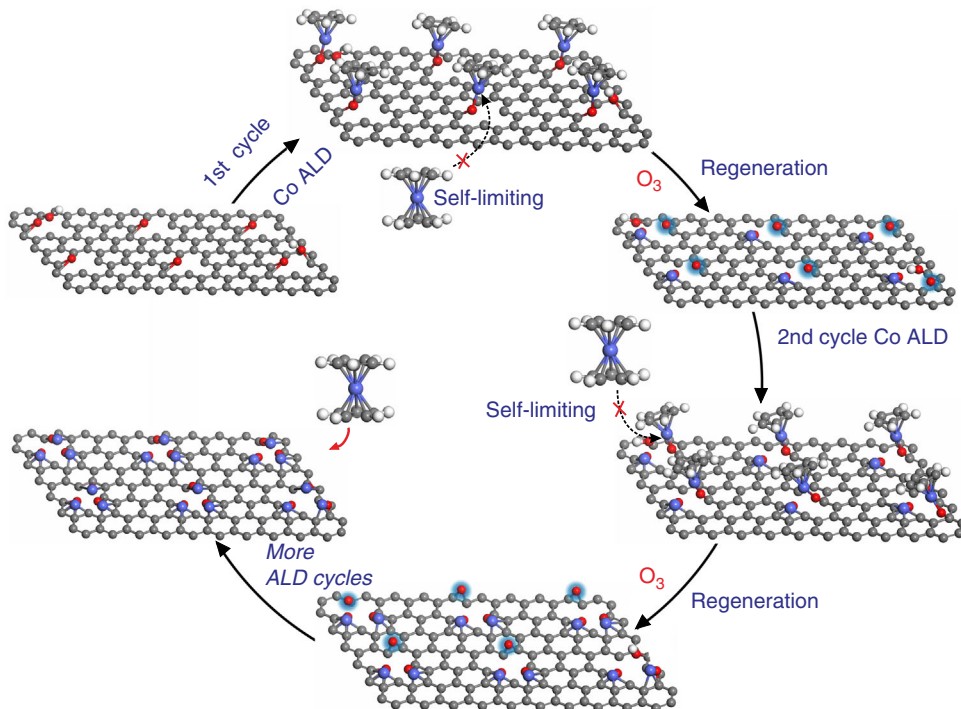

**Fig. 1** Schematic illustration of the synthesis of $Co_1$/G SACs with tuneable loadings. The first cycle of Co ALD by an alternative exposure of the support to $CoCp_2$ vapor and $O_3$ gas at 150 °C; the second cycle of Co ALD on $Co_1$/G to deposit another batch of Co atoms on the active sites created by $O_3$ treatment at 150 °C in the previous Co ALD cycle; more cycles of Co ALD results in a high loading of $Co_1$ SACs. The balls in gray, white, red, and blue represent carbon, hydrogen, oxygen, and cobalt, respectively

into particles or other Co species (Supplementary Figure 5a-5d and Supplementary Figure 6). The presence of isolated Co atoms was further confirmed by EDS-mapping (Supplementary Figure 5e–g). Interestingly, an increase in the Co loading generated by performing more cycles of Co ALD continued to produce well-dispersed Co single atoms rather than large Co clusters and nanoparticles (Fig. 2c–e, Supplementary Figure 7–9). To our delight, the Co particles or clusters were barely present on graphene even at high loading densities of 2 wt% and 2.5 wt% (Fig. 2d, e, Supplementary Figure 8, 9). We also found that the density of Co single atoms loaded on graphene is closely correlated to the amount of epoxy groups present on the support (Fig. 2g), which further supports the idea that the epoxy groups act as anchor sites for the Co precursors as illustrated in Fig. 1. In order to probe the local chemical environment of Co atoms, we have conducted spatial-dependent electron energy loss spectra (EELS) measurements off (**1**) and on (**2**, **3**) Co atom sites as marked in the inset of Fig. 2f. In contrast to the featureless curves (green) taken in the bare graphene region, the EELS acquired on single Co atom sites reveal the coexistence of Co $L_{2,3}$ edge and O K edge related peaks, suggesting that the anchoring of isolated Co atoms in graphene involves oxygen atoms. Furthermore, examination of the Co $L_{2,3}$ edge fine structure shows sharp white features with an $L_3/L_2$ ratio of ~5, suggesting an oxidation state which is lower than +2 valence state[37]. Such an atomic insight not only provides compelling evidence for the presence of Co–O bonds at the catalytically active sites but also rationalises the proposed atomic structures of $Co_1$/G SACs as will be discussed in more details later.

The XAFS is a state-of-the-art method to probe the local information of the adsorbing atoms. In our experiment, it was used to investigate the structural and electronic states of the $Co_1$/G SACs with different loadings. As shown in Fig. 3a, the XANES white line peaks of the $Co_1$/G SACs samples with different

loadings are centered at 7727.1 eV, between that of the Co foil (7725.7 eV) and $Co_3O_4$ (7728.0 eV), consistent with the $Co_1$/G SACs as-prepared being in the oxidized state rather than the metallic state. Additional structural information can also be explicitly inferred from the extended X-ray adsorption fine structure (EXAFS) spectra at the Co K-edge (Fig. 3c and Supplementary Figure 1, 0). Further, the Fourier transform (FT) $k^3\chi(k)$ spectrum of the $CoCp_2$ molecule exhibits a dominant peak centered at 1.60 Å assignable to the Co–C bonds of the $CoCp_2$ precursors. In contrast, the EXAFS spectrum (labeled as CoCp/G) acquired on the graphene support after exposure to $CoCp_2$ vapor at 150 °C shows one main peak at 1.58 Å. This suggests the existence of a shorter bond which may result from the chemisorption of $CoCp_2$ precursors to graphene. As illustrated in Fig. 1, it is naturally expected that $CoCp_2$ precursors react with the epoxy groups on graphene by removing one of Cp ligands. Hence, the resultant Co atoms will be bonded to one remaining Cp ligand and oxygen atoms on graphene, giving rise to a shorter Co–C/O bond length as compared with that of Co–C bonds in $CoCp_2$ molecules. In addition, the FT spectra for a series of $Co_1$/G SACs show that the first shell peaks undergo a further downshift to 1.56 Å as compared to that of CoCp/G, indicating that the bonding of Co on the basal plane of graphene is further strengthened due to the Co atoms forming new chemical bonds with graphene after the complete removal of organic ligands via the $O_3$ treatment. These observations are consistent with the bonding information extracted from the EXAFS fitting results (Supplementary Table 1). It is also worth noting that the major peaks at 2.18 Å and 2.48 Å of the FT spectra acquired on Co foil and $Co_3O_4$ respectively are absent in the corresponding spectra of all the $Co_1$/G SACs, which further confirms that Co atoms remained well dispersed on the graphene support at the high loading of 2.5 wt%, in line with the STEM results.

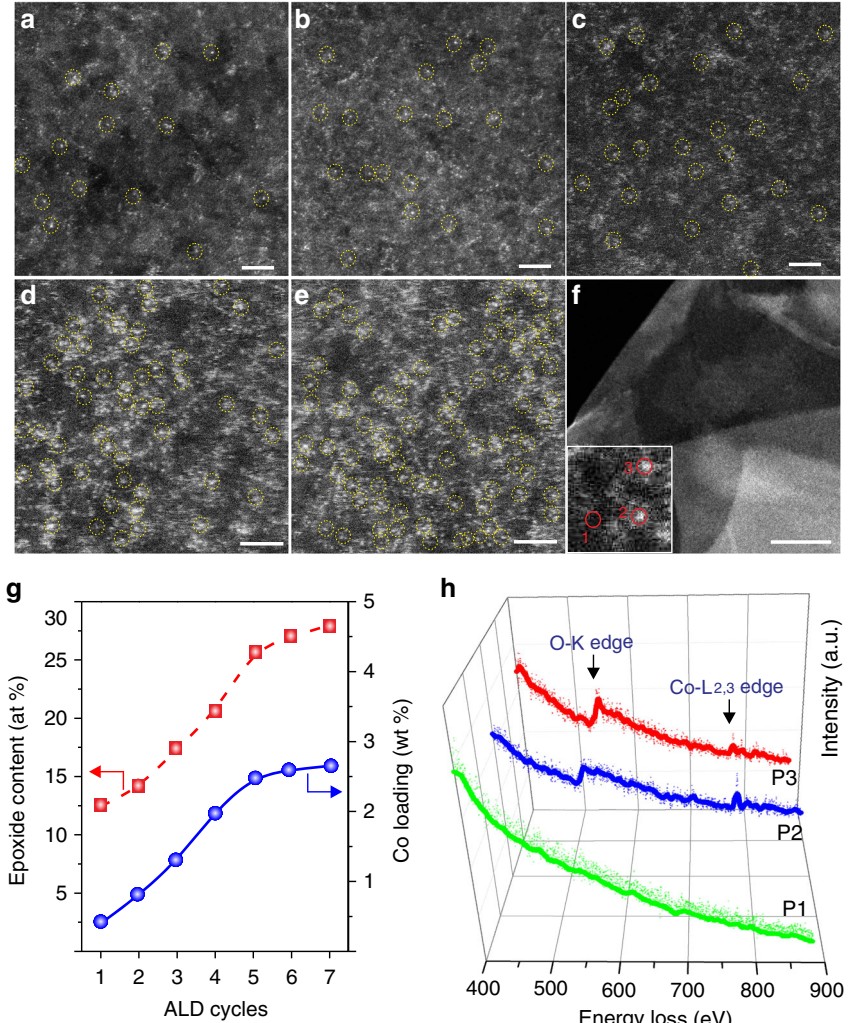

**Fig. 2** Structural characterization and identification of Co$_1$/G SACs. Aberration-corrected STEM-ADF images of Co$_1$/G-0.4 (**a**), Co$_1$/G-0.8 (**b**), Co$_1$/G-1.3 (**c**), Co$_1$/G-2.0 (**d**), and Co$_1$/G-2.5 (**e**). Scale bars, 2 nm (**a–e**), 50 nm (**f**). Co$_1$ single atoms are highlighted by yellow dashed circles. **f** The STEM-ADF image of Co$_1$/G-2.5 catalysts at low magnification. **g** The evolution of epoxy content in graphene and Co loadings of Co$_1$/G SACs catalysts as a function of the number of ALD cycles. **h** EEL spectra of O K-edge and Co L$_{2,3}$-edge acquired in the bare graphene region (position 1 as marked in the inset of **f**) and the isolated Co atom sites (Position 2, 3 as marked in the inset of **f**)

**DFT calculations**. In order to determine the atomic structures of the Co$_1$/G SACs, we performed DFT calculations in combination with a standard XAFS fitting method (Supplementary Figure 1, 2-1, 7). Based on the plausible Co ALD reaction mechanism on the defect-rich graphene support (Supplementary Figure 1, 1), it is most likely that the isolated Co atoms are anchored in vacancy related structures through bonding to oxygen species as revealed in our EELS measurements. We hence propose several possible atomic configurations of the Co$_1$/G SACs along this line (Supplementary Figure 1, 2-1, 3), which are further optimized via DFT calculations. Our calculations reveal a stable structure consisting of CoCp bonded to the graphene support via two interfacial O atoms and one C atom for the CoCp/G sample (Supplementary Figure 1, 2a). Such a structure is expected to be generated in the first step of ALD cycle (Fig. 1), in line with previous work[38]. After the removal of ligand (Cp), isolated Co atoms in the Co$_1$/G SACs can be anchored to the divacancy of graphene through bonding with two interfacial O atoms and four C atoms, forming a new structure represented by Co$_1$–O$_2$C$_4$ (Supplementary Figure 1, 2b). In order to verify these two structures, we calculated their XANES (Fig. 3b) and fitted their EXAFS (Supplementary Figure 1, 5-1, 7) spectra, which show a good agreement with our experimental

data acquired on CoCp/G and Co$_1$/G SACs respectively (Fig. 3b, Supplementary Figure 1, 3-1, 7 and Supplement Table 1). In contrast, the calculated XANES spectra of other DFT-modeled structures fail to reproduce the main features of experimental curves (Supplementary Figure 1, 4). Hence, it is most likely the Co$_1$/G SACs prepared here contains a six-coordinated structure (Co$_1$–O$_2$C$_4$), wherein individual Co atoms are bonded to two interfacial O atoms and four C atoms.

**Catalytic activity**. Azoxybenzene is one of the most important industrial media of the dye and pharmacy industries[39,40]. The catalytic selective hydrogenation of nitrobenzene has been a major method for the synthesis of azoxybenzene. Unfortunately, the catalysts employed for industrial-scale production of azoxybenzene are usually toxic[41]. Noble metal heterogeneous catalysts have recently emerged as promising catalysts for the synthesis of azoxybenzene[42–44]. However, these catalysts are not practical for the scalable synthesis of azoxybenzene due to their high cost. Here we employ the Co$_1$/G SACs in the selective hydrogenation of a wide range of substituted nitrobenzene to produce azoxy products as illustrated in Fig. 4a. As shown in Fig. 4b,

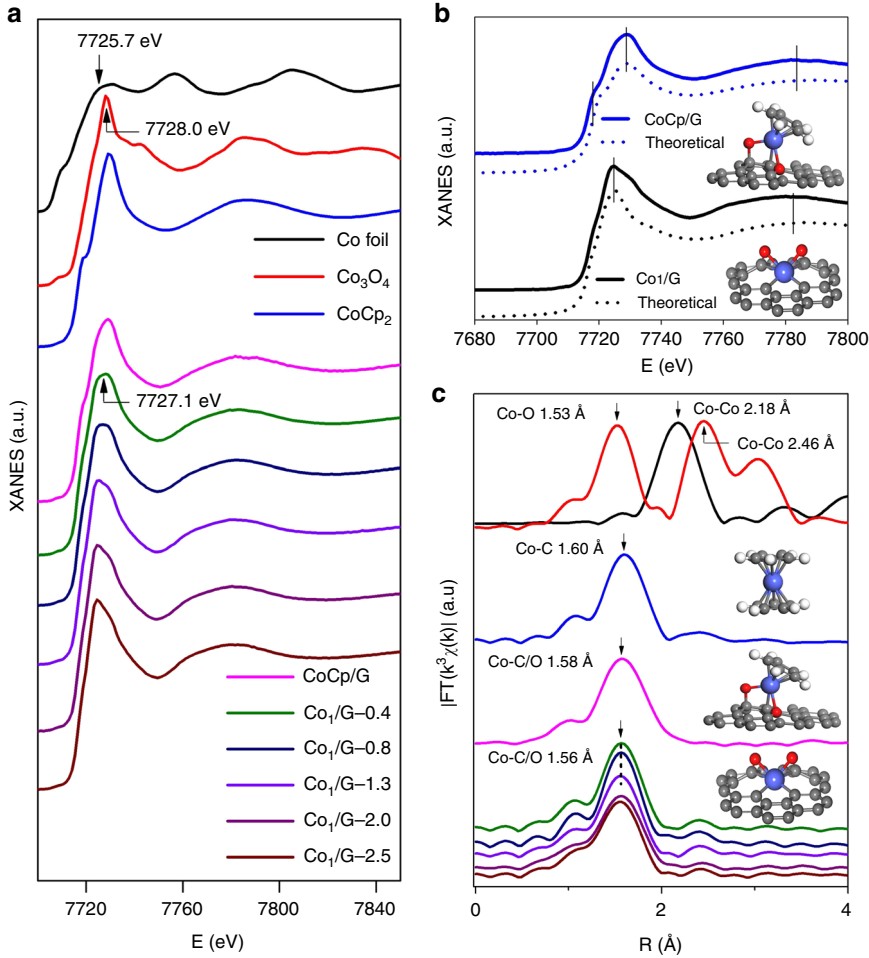

**Fig. 3** Co K-edge XAFS and EXAFS spectra of CoCp/G, Co$_1$/G SACs. **a** Co K-edge XAFS spectra. **b** The experimental XANES curves are compared with the calculated XANES data of optimized DFT-modeled structures of Co$_1$/G and CoCp/G (inset shows the atomic structures of the models). **c** Fourier transform (FT) extended x-ray absorption fine structure (EXAFS) of these samples with the corresponding structures (insets). The balls in gray, white, red, and blue represent carbon, hydrogen, oxygen, and cobalt, respectively. The Co K-edge XAFS and EXAFS spectra of Co foil, Co$_3$O$_4$ and CoCp$_2$ samples are displayed for comparison. Note: the figure legend in **a** also applies to **c**

all the Co$_1$/G SACs exhibit much higher selectivity to 3, 3'-dichlorideazoxybenzene (98%), compared to Pt/carbon (18%) and Co-NPs/G (4%) (Supplementary Figure 1, 8-1, 9). For different substituted nitrobenzene (compound 1–6) (Fig. 4c, d and Supplementary Figure 20), all the Co$_1$/G SACs also exhibit significantly higher selectivity to azoxy compounds (~90%) compared to Pt/carbon (18~21%) and Co-NPs/G (2-3%). The major product over Pt/carbon and Co-NPs/G catalysts is aniline compound, giving rise to a low selectivity to azoxy compounds[45–47]. In addition, the Co nanoparticles (Supplementary Figure 2, 1) synthesized by ALD (designated as Co-NPs/G-ALD) show a low selectivity (~5%) to azoxy compounds (Supplementary Figure 22). All the azoxybenzene products were verified by their characteristic $^1$HNMR and $^{13}$CNMR spectra (Supplementary Figure 23-36)[48]. Importantly, the Co$_1$/G SACs with different loadings show negligible variations in the selectivity towards all azoxy compounds (Fig. 4b–d), presumably due to a relatively uniform dispersion of isolated Co atoms for all the Co$_1$/G SACs catalysts.

The excellent atomic dispersion of all the Co$_1$/G SACs with different loadings indeed results in a similar turnover frequency (TOF) of 0.33 s$^{-1}$, which is 6 times higher than that of Co nanoparticles (0.05 s$^{-1}$) as shown in Fig. 4e. In addition, Co$_1$/G SACs synthesized exhibit higher catalytic activity and selectivity in the hydrogenation of nitrobenzene as compared to non-noble

metal catalysts reported in the previous work (Supplementary Table 2). Moreover, as shown in Fig. 4e, the TOF of Co$_1$/G SACs is even higher than that of Pt/carbon (0.23 s$^{-1}$), which proves their superior catalytic performance comparable to that of precious catalysts. To our delight, Co$_1$/G SACs with a high loading of 2.5 wt% also exhibit a high durability in the selective hydrogenation of nitroarenes as revealed in the recyclability test (Supplementary Figure 37).

The hydrogenation of nitrobenzene is expected to occur through multiple steps, involving the generation of different reaction intermediates (Supplementary Figure 38)[49,50]. The adsorption energy of the intermediate compounds on catalytic surface is one of the key factors that determine the selectivity to certain target products[51]. In our system, we observed that azoxy compound is the major product when Co$_1$/G SACs is applied. This indicates that the reaction stops at step 4 (Supplementary Figure 38) is prohibited, preventing the further hydrogenation of azoxy compound to aniline. Such a reasoning is also reported in the previous work[52]. Therefore, we performed DFT calculations of the adsorption energies of azoxybenzene molecules on Co (111), Co$_1$/G SACs and catalytic centers consisting of isolated Co atoms coordinated to four carbon atoms in a graphene divacancy (labeled as Co$_1$-C$_4$/G). It's worth mentioning here that the catalytic activity of graphene-based metal SACs has been

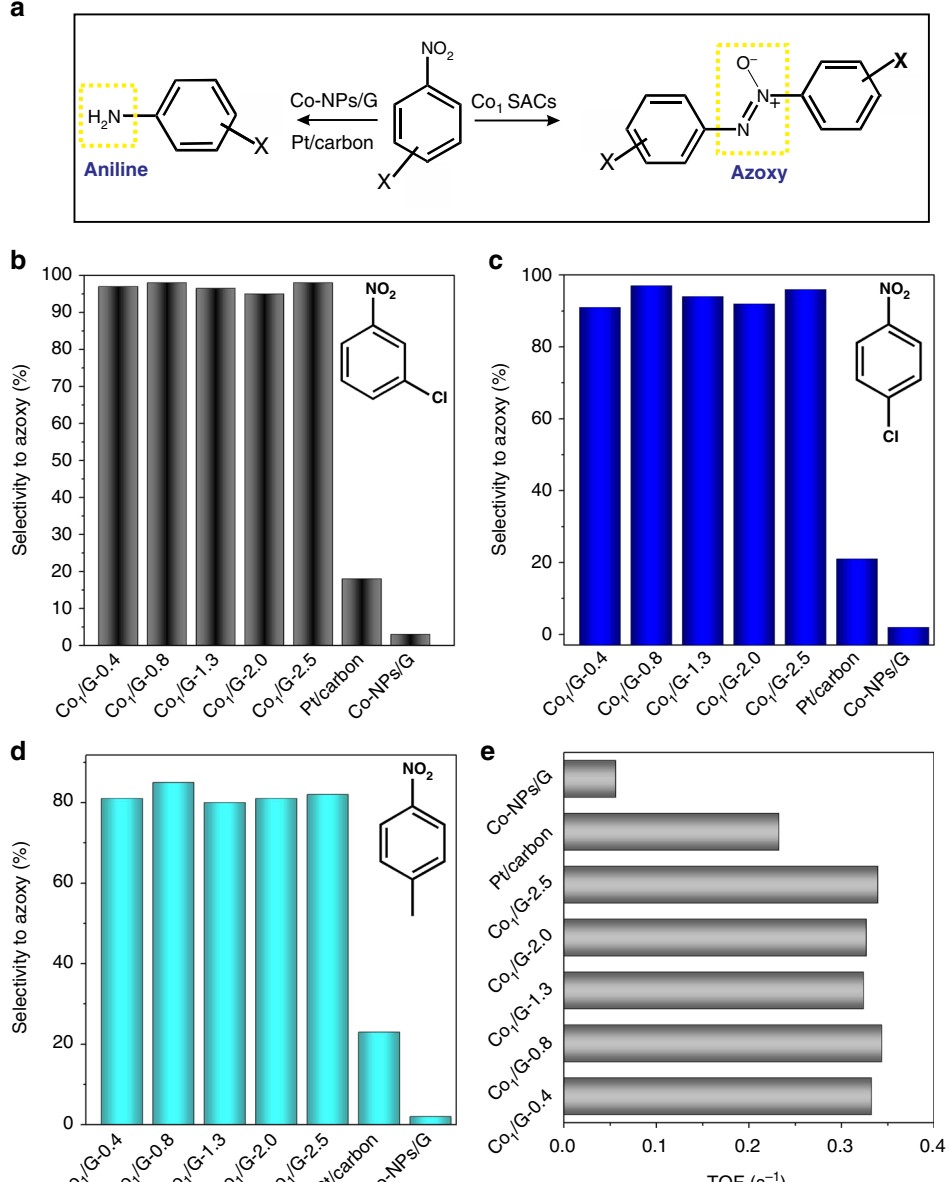

**Fig. 4** Catalytic selectivity to azoxy products of various Co$_1$/G SACs catalysts. **a** Schematic illustration of the hydrogenation of nitroarenes using different catalysts. Histograms of the selectivity to azoxy products for the hydrogenation of 1-chloride-3-nitrobenzene (**b**), 1-chloride-4-nitrobenzene (**c**), and 1-methyl-4-nitrobenzene (**d**) at ~100% conversion of nitroarenes by using different Co$_1$/G SACs including Co$_1$/G-0.4, Co$_1$/G-0.8, Co$_1$/G-1.3, Co$_1$/G-2.0, Co$_1$/G-2.5, Co-NPs/G, and Pt/carbon. **e** Turnover frequency (TOF) of the different catalysts tested in the selective hydrogenation of nitrobenzene

predicted in previous theoretical studies[53–55], but catalytic role of atoms proximal to single metal atom remains largely unexplored. Here, we found that holding Co and oxygen atoms in the close proximity is the key that allows the reaction to proceed with extremely high selectivity in the partial hydrogenation of nitrobenzene to azoxybenzene. Such an excellent catalytic performance can be attributed to the different binding nature of the reactants adsorbed at different catalytic sites. As shown in the Fig. 5a, the dispersion-corrected DFT (DFT-D2) calculations revealed that azoxybenzene shows a stand-up adsorption geometry on Co (111) with a large adsorption energy of −1.15 eV. In the case of the Co$_1$–C$_4$/G, the azoxybenzene exhibits a flat adsorption geometry over the Co$_1$–C$_4$ site with an even larger adsorption energy of −1.55 eV (Fig. 5a). In contrast, the azoxybenzene binds to the Co$_1$–O$_2$C$_4$ site of Co$_1$/G SACs weakly with a small adsorption energy of 0.52 eV. In addition, the

separation between azoxybenzene and the Co catalytic center becomes larger for the weak adsorption case (Co$_1$/G SACs) (Supplementary Figure 39). Furthermore, it is observed that the charge redistribution at the interface in the cases of Co (111) and Co$_1$–C$_4$/G is more significant than that of azoxybenzene adsorbed on the Co–O$_2$C$_4$ site of Co$_1$/G SACs (Supplementary Figure 40a-c). The weak adsorption of azoxy compounds over Co$_1$/G SACs might be insufficient to break the N–O bond of azoxybenzene for further hydrogenation, giving rise to a high selectivity to azoxybenzene (Supplementary Figure 40d)[49,52].

To gain more insights into the catalytic role of proximal atoms in the SACs, we also calculated the projected density of states (PDOS) of the Co atom to *d* and *s* orbitals for both Co$_1$/G and Co$_1$–C$_4$/G (a hypothetic structure without proximal oxygen atom for a comparison). As shown in Fig. 5b, the PDOS of Co 3d orbitals around Fermi energy (E$_F$) is dramatically different in the

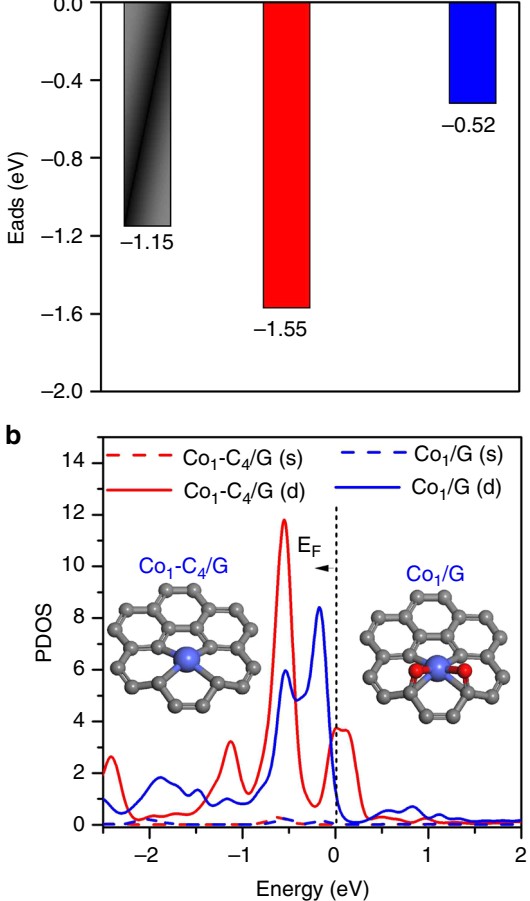

**Fig. 5** Theoretical simulations of the catalytic origins. **a** Adsorption energies for the azoxybenzene on Co (111) facet, $Co_1-C_4/G$ and $Co_1/G$ SACs. **b** The partial density of state (PDOS) projected on the Co 4 s and 3d orbitals of $Co_1-C_4/G$ and $Co_1/G$. The balls in gray, red, and blue represent carbon, oxygen, and cobalt, respectively

two examined structures while the difference for PDOS of Co 4 s orbitals is much less significant. The PDOS of Co 3d of $Co_1–C_4/G$ exhibit a noticeable peak at $E_F$ contributed by the partially filled d orbitals. In contrast, the presence of oxygen atoms proximal to $Co_1$ in $Co_1/G$ SACs pushes these partially filled d orbitals above $E_F$, resulting in a lower PDOS at $E_F$ and thus a more positively charged Co atom. Consistently, Bader charge transfer analysis[56] also reveals that each Co atom in $Co_1/G$ SACs loses more electrons (~0.8 electrons) to the surrounding O and C atoms compared to Co atoms in $Co_1–C_4/G$ (lose 0.65 electrons to the surrounding atoms). A more positively charged $Co_1–O_2C_4$ active center disfavors the adsorption of electron-deficient azoxybenzene on the $Co_1–O_2C_4$ of $Co_1/G$ SACs for the further hydrogenation to azobenzene, which in turn leads to a higher selectivity towards azoxybenzene (Supplementary Figure 40d)[51,57]. The excellent catalytic performance of $Co_1/G$ SACs discovered in our study attests to their great potential in a wide range of selective hydrogenation reactions.

In conclusion, we have developed a stepwise approach to fabricate a series of $Co_1/G$ SACs with high and precisely tunable loadings. Our results reveal that the ozone treatment of a graphene support at mild ALD conditions not only burns off metal ligands, but also recreates active sites for the subsequent anchoring of another batch of Co atoms. This unique approach allows us to precisely tune the density of the supported Co atoms from 0.4% up to 2.5% without formation of any Co nanoparticle

or clusters. As compared to conventional Co nanoparticles and precious Pt /carbon catalysts, all the $Co_1/G$ SACs exhibit remarkably high selectivity towards azoxy compounds in the hydrogenation of nitrobenzene aromatics. This can be attributed to the electronic coupling between Co atoms and adjacent oxygen atoms that results in a positively charged catalytic center. Consequently, the adsorption of electron deficient azoxy compounds is weaker and thus the full hydrogenation of nitroarenes is prevented. Our findings have opened up an unprecedented avenue to precisely control the loading of single metal atoms in a wide range of SACs for industrially important chemical transformations.

## Methods

**Materials**. All the chemicals were purchased from Sigma Aldrich and were used as received without further purification. These includes Bis (cyclopentadienyl) cobalt ($CoCp_2$, 98%), $Co(NO_3)_2·6H_2O$ (98%, trace metals basis), the Pt/carbon catalyst, sodium borohydride (99.99%, trace metals basis) and all the subsitituted nitrobenzenes . Few-layer graphene oxide and pristine graphene nanosheet (99.5%) were purchased from Nanjing XFNANO Materials Tech Co. Ltd. and Chengdu Organic Chemicals Co. Ltd., Chinese Academy of Sciences respectively.

**Synthesis of $Co_1/G$ SACs**. The synthesis of $Co_1$ SACs was performed in a viscous ALD flow reactor (Plasma-assisted ALD system, Wuxi MNT Micro and Nanotech Co., Ltd, China) by alternatively exposing thermally-reduced graphene oxide to $CoCp_2$ precursor and $O_3$ at 150 °C. Ultrahigh purity $N_2$ (99.99%) was used as carrier gas with a flow rate of 50 mL/min. The Co precursor was heated at 100 °C to generate a high enough vapor pressure. The reactor and reactor inlets were held at 150 °C and 120 °C respectively to avoid any precursor condensation. An in-situ thermal reduction of as-received graphene oxide support was conducted at 300 °C for 5 min before performing Co ALD. The timing sequence was 100, 120, 150, and 120 seconds for the $CoCp_2$ exposure, $N_2$ purge, $O_3$ exposure and $N_2$ purge, respectively. Conducting Co ALD with 1, 2, 3, 4 and 5 cycles allows for the synthesis of $Co_1/G$-0.4, $Co_1/G$-0.8, $Co_1/G$-1.3, $Co_1/G$-2.0, and $Co_1/G$-2.5, respectively.

**Synthesis of Co-NPs/G**. The Co-NPs/G was synthesized using the previous method[58]. In brief, 100 mg of the graphene oxide was dispersed in 15 mL of ethanol with sonication. Meanwhile, 10 mg of $Co(NO_3)_2·6H_2O$ (1 mmol) was added into 15 mL of ultrapure water. After that, NaOH solution (6 M) was added into as-prepared $Co(NO_3)_2$ solution. $Co(OH)_2$ precipitated was filtrated and washed multiple times using ultrapure water and ethanol. The $Co(OH)_2$ solid was then dispersed into ethanol with sonication, and gradually added into graphene oxide dispersion followed by a continuous stirring for 1 h. In order to keep the pH at 11, 6 M NaOH solution was used during the synthesis. Then, 1 ml of $N_2H_4·H_2O$ was added and stirred for another 30 min. The obtained mixture was put into Teflon-lined stainless autoclave at 180 °C for 12 h. The sample was dried at 60 °C after filtration and washing. Before used in the catalytic reaction, as-prepared samples were calcined at 150 °C in air, and reduced under a flow of 10% $H_2/Ar$ gas at 150 °C.

**The characterization of as-prepared catalysts**. The Co loadings in all the samples were measured by an inductively coupled plasma atomic emission spectrometer (ICP-AES); therein all samples were dissolved in hot fresh aqua regia. XPS measurements were carried out in a custom-designed ultrahigh vacuum (UHV) system with a base pressure better than $2 × 10^{-10}$ mbar. Al Ka ($hv$ = 1486.7 eV) was used as the excitation sources for XPS. [1]H and [13]C NMR spectra were recorded on a Bruker AV 300 (300 MHz) and Bruker AV500 (500 MHz) spectrometer. Chemical shifts were reported in parts per million (ppm), and the residual solvent peak was used as an internal reference: [1]H (chloroform δ 7.27), [13]C (chloroform δ 77.0).

STEM-ADF characterization and image simulation: STEM-ADF imaging was carried out in an aberration-corrected JEOL ARM-200F system equipped with a cold field emission gun and an ASCOR probe corrector at 60 kV. The images were collected with a half-angle range from ~85 to 280 mrad, and the convergence semiangle was set at ~30 mrad. The imaging dose rate for single frame imaging is estimated as $8 × 10^5$ e/nm²·s with a total dose of $1.6 × 10^7$ e/nm². The dwell time for STEM-ADF image is set as 20 us/pixel. The EELS 2D maps were taken by Gatan Quantum ER Spectrometer with a spectrum pixel time of 2 s.

The X-ray absorption near edge structure (XANES) and the extended X-ray absorption fine structure (EXAFS) measurements of Co K-edge were carried out at the XAFCA beamline of the Singapore Synchrotron Light Source (SSLS)[59]. The storage ring of SSLS operated at 700 MeV with beam current of 250 mA. A Si (111) double-crystal monochromator was applied to filter the X-ray beam. Co foils were used for the energy calibration, and all samples were measured under transmission

mode at room temperature. The EXAFS oscillations χ(k) were extracted and analyzed using the Demeter software package[60].

**DFT calculations**. The first-principles calculations are performed with density functional theory (DFT) by utilizing the Vienna ab-initio Simulation Package (VASP)[61]. The generalized gradient approximation (GGA) in the Perdew-Burke-Ernzerh (PBE) format[62,63] and the projector-augmented wave (PAW) method[64] are employed in all calculations. A plane wave basis with a cut-off energy of 450 eV along with spin polarization and $4 \times 4 \times 1$ k-sampling in Brillouin zone are used for all calculations. The convergence criterion for structural relaxations is set to 0.01 eV/Å. Effects of Van der Waals force (through DFT + D2)[65] are also considered. Defective graphene is defined with an $8 \times 8$ unit cell with a divacancy consisting of two missing adjacent C atoms (Figure S5). The Co (111) surface is described with a $4 \times 4$ unit cell with 5 atomic layers. The Co atoms of the bottom three layers are kept fixed in the relaxation process. 18 Å of a vacuum layer in perpendicular direction is included into supercells to avoid unphysical interactions between neighboring unit cells.

**XANES simulations**. The XANES spectra of Co K edges of all the structures predicted by DFT were modeled using a finite difference method implemented by FDMNES program[66]. For the FDMNES calculation, the Schrödinger equation is solved with a free shape of potential, which avoids using the muffin-tin approximation and thus better reproduce the theoretical XANES spectrum. For all the calculated XANESs, the final states are calculated inside a sphere with a size of 8 Å. The energy step at the Fermi level is 0.2 eV.

**Hydrogenation of nitrobenzene and its derivatives**. 0.1 mmol of substrates, $1.4 \times 10^{-3}$ mmol of catalyst, 2 mmol sodium borohydride, 8 mL of Tetrahydrofuran (THF) and 2 ml of ultrapure water were mixed in a round-bottom flask to carry out the reaction. Then the reaction mixture was stirred at 25 °C for one hour. The as-obtained products were analyzed by GC-MS (7890 A GC system, 5975 C inert MSD with Triple-Axis Detector, Agilent Technologies).

**Data availability**. The data that support the findings of this study are available from the corresponding author upon reasonable request.

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

## Acknowledgements

J. Lu acknowledges the support from NUS start-up grant (R-143-000-621-133) and Tier 1 (R-143-000-637-112) and MOE Tier 2 grant (R-143-000-A06-112). W. Chen acknowledges NSFC grant 91645102. C. Su acknowledges NNSFC (51502174) and Shenzhen Peacock Plan (Grant No. 827-000113, KQTD2016053112042971). C. Zhang acknowledges the support from Singapore National Research Foundation (NRF-CRP13-2014-03 and NRF-CRP16-2015-02). H. Yan appreciates the funding support from the China Postdoctoral Science Foundation (2017M610541).

## Author contributions

J.L. supervised the project. H.Y. and J.L. conceived and designed the experiment. H.Y. prepared the catalysts and performed the activity test with the supervision of C.S. and J. L.; X.Z. performed the STEM-ADF and EELS characterization with the supervision of S.J. P.; N.G. performed the theoretical calculation under the supervision of C.Z.; Y.D. and S. X. helped to perform the XAFS measurement. Z.L. helped to establish the ALD system under the supervision of W.C.; R.G. performed the XPS measurements under the supervision of W.C.; C.C. helped to perform the TEM characterization. Z.C., W.L., C.Y., and J.Li. assisted in the activity test. H.Y., C.S., C.Z., and J.L. wrote the manuscript. All authors reviewed and contributed to the manuscript.

## Additional information

**Competing interests:** The authors declare no competing interests.

