## [Peer Review File · Nature Communications]

Reviewers' comments:

Reviewer #1 (Remarks to the Author):

In this work the authors showed that atomically dispersed cobalt atoms can be fabricated on graphene with a tunable high Co loading up to 2.5 wt% using atomic layer deposition (ALD). During CoO oxide ALD, the ozone exposure time generate additional nucleation sites, playing the key role for achievement of the high Co loading. In hydrogenation of nitroarenes under mild reaction conditions, the Co1 single-atom catalyst showed exceptional activity and high selectivity to azoxy aromatic compounds, compared to conventional Co nanoparticles and Pt/carbon catalysts. The Co single atom catalysts also showed good stability after five recycles. They further carried out DFT calculations to compare the adsorption energies of azoxybenzene molecules on the different Co species, and suggested that the oxygen proximity atoms play the key role for the activity enhancement. It is very unusual that Co single-atom catalyst showed even higher activity than Pt noble catalyst. The work is highly recommended for publication in Nature communications, while the following issues has to be addressed:

1. Comparing the activity of Co1/G single-atom ALD catalysts with a Co-NPs catalyst synthesized by wet-chemistry might not be that straightforward, since the method of catalyst synthesis often plays a considerable role in catalyst activity. What happens to the activity if the Co1 single-atoms aggregated to clusters after treating the Co1/G at elevated temperatures?
2. The close correlation of Co loadings with the amount of epoxy groups Figure 2g might not be able to support that the epoxy groups act as anchor sites for the Co precursors as illustrated, since the O showed in XPS might come from the oxidation of Co by ozone, rather than the formation of epoxy groups. The authors might like to explain it.
3. Hydrogen dissociation in single-atom catalyst might be the rate-determining step, the authors might also like shows the ability of hydrogen dissociation on Co1 single-atoms.

Other minor issues:

1. On page 20, line 612, “orbitals of Co1-C4/G and Co1-C4/G.” Is there a mistake showing two times of Co1-C4/G?
2. On page 8, line 259, the first appearance of “DFT+D2” in the text is not defined.
3. In Fig. 1. The front color of “regeneration” is not very clear, it is better to change another color.
4. In Fig. 5b, the structures of the two insets should be clearly defined.

Reviewer #2 (Remarks to the Author):

In this manuscript the authors employed atomic layer deposition (ALD) technique to prepare a series of Co-based single-atom catalysts (SACs) with different metal loading via controlling the number of Co ALD cycles, and investigated the catalysts for hydrogenation of nitrobenzene to azoxybenzene. The structure of CoO₂C₄ was determined with EXAFS fitting, and this Co1/G SACs showed 68% selectivity to azoxybenzene in nitroarenes reduction using sodium borohydride (NaBH₄) as a reducing agent. DFT calculation results proposed positively charged Co1 active centre disfavoured the adsorption of electron-deficient azoxybenzene which resulted in a higher selectivity towards azoxybenzene. My overall impression is that the paper contains rigorous material characterizations but lacks conceptual advance and novelty that are required for publication in high-level journal like Nature Communications. Below are major concerns:

1. There have been reports on using ALD technique to prepare SACs. For example, Pt₂/graphene (ref.17), Pt₁/graphene (ref.15) and others. In fact, ALD has been proved a powerful tool for synthesis of stable single-atom or even cluster catalysts. Since ALD has the ability to precisely control the size and distribution of particles on a substrate by using sequential and self-limiting surface reactions, it is not surprising that SACs with different loading can be obtained by precisely controlling the number of Co ALD cycles.
2. Authors proposed the atomic structure of Co centre in Co1/G SAC was CoO₂C₄ moieties by comparing the bonding length of EXAFS fitting parameters to those in several DFT models. This method for determining the atomic structures could be ambiguous. To confirm the structure, XANES should be used and it is more sensitive to the geometrical arrangement of atoms around the centre atom (Nat. Mater., 2015, 14, 937; Nat. Commun., 2017, 8, 957; ref.30). XANES simulation of those possible DFT models to further determine the structural configuration should be more acceptable.
3. The catalytic hydrogenation tests was performed by using sodium borohydride (NaBH₄) as a reducing reagent. Green reducing reagent like H₂ molecule is more interesting. And 68% selectivity to azoxybenzene was not high, many catalysts systems can reach above 90% selectivity (J.Catal., 2016, 336, 41; Green Chem., 2016, 18, 3852, and so on).
4. The substrate scope should be widely extended by using different functional and structurally diverse substrates.

Reviewer #3 (Remarks to the Author):

The study deals with a new preparation method for highly dense single atom catalysts and testing these materials in a relevant reaction with important applications. I find the material synthesis

and the experimental work excellent and certainly I am glad to support the work for publication. However, I have major concerns on the computational study part which I find below the state-of-the-art.

1) The models employed are speculative and not supported by any of the experimental results. A closest match between experiment and theory is needed, otherwise the computational part can be removed without the article losing any valuable insight.

2) Out of the different O-containing functional groups only one is the epoxy with a neighboring empty cavity. As epoxides can directly exist on graphene the choice seems rather arbitrary.

3) The fact that the turn-over frequency is just 6 times higher than that of nanoparticles implies that most of the Co atoms are actually non active. Can the coordination be inspected prior to use to see how much Co is trapped in the precursor form.

4) The active phase can be derived from computational calculations, however the paths are not presented. The work will benefit from running the whole reaction profiles.

5) The reason for selectivity is rather speculated than inferred from the data. Either the full reaction path is calculated on the proposed structure or the claims need to be toned down.

6) D2 is not correct to deal with metal surfaces. The results are overbinding by a significant amount.

7) The description in terms of density delocalization for a van der Waals interaction is very misleading.

8) The units for electron charge are $|e^-|$ and not e

9) The functional is misspelled.

10) 400 eV cutoff is below the VASP recommendation (415-450 eV for O)

11) The origin of the divacancy in graphene is not justified.

12) The use of the same k-point sampling in all the calculations is wrong.

13) Figure 5a E_a stands for activation energy and not adsorption energy usually noted as E_{ads}

14) The insets in Figure 5b need to be assigned in the figure.

15) Figure 5 c-f do not represent any relevant information for the discussion they shall go to the Supplementary documentation.

16) The description in terms of the volcano for the data presented in the study is completely irrelevant as no model for the optimum value of the adsorption is considered. The authors need to reevaluate this part.

Minor issues

1) The hydrogenation of nitroarenes has been reported to occur for Pt NanoSelect particles, at least a reference to this work and a comparison is needed.

2) The authors are not considering carbon nitride scaffolds for which 1 % weight can be also obtained due to entropic contributions.

Point-by-point response letter

Reviewers' comments:

Reviewer #1 (Remarks to the Author):

In this work the authors showed that atomically dispersed cobalt atoms can be fabricated on graphene with a tunable high Co loading up to 2.5 wt% using atomic layer deposition (ALD). During CoO oxide ALD, the ozone exposure time generate additional nucleation sites, playing the key role for achievement of the high Co loading. In hydrogenation of nitroarenes under mild reaction conditions, the Co₁ single-atom catalyst showed exceptional activity and high selectivity to azoxy aromatic compounds, compared to conventional Co nanoparticles and Pt/carbon catalysts. The Co single atom catalysts also showed good stability after five recycles. They further carried out DFT calculations to compare the adsorption energies of azoxybenzene molecules on the different Co species, and suggested that the oxygen proximity atoms play the key role for the activity enhancement. It is very unusual that Co single-atom catalyst showed even higher activity than Pt noble catalyst. The work is highly recommended for publication in Nature communications, while the following issues has to be addressed:

Response: We appreciate the reviewer's comments and support for the publication of our work.

1. Comparing the activity of Co₁/G single-atom ALD catalysts with a Co-NPs catalyst synthesized by wet-chemistry might not be that straightforward, since the method of catalyst synthesis often plays a considerable role in catalyst activity. What happens to the activity if the Co₁ single-atoms aggregated to clusters after treating the Co₁/G at elevated temperatures?

Response: We appreciate the reviewer's valuable comments.

In view of referee's comments, we have prepared Co nanoparticle catalysts using ALD (designated as Co-NPs/G-ALD) followed by thermal annealing of Co₁/G SACs at 450 °C for 1 h. As shown in **Figure R1**, the average size of Co particle is determined to be ~4.3 nm. The catalytic performance of Co-NPs/G-ALD was also

evaluated for the hydrogenation of nitrobenzene. Our new experimental data (**Figure R2**) shows that the Co-NPs/G-ALD exhibits a slightly higher selectivity (~5%) to azoxy compounds compared to Co-NPs/G prepared by wet chemistry (**Figure 4**). Both Co nanoparticles catalysts prepared by ALD and wet-chemistry methods demonstrate significantly lower selectivity to azoxy compounds compared to that of Co₁/G SACs.

We have included this sentence in the revised manuscript: “**In addition, the Co nanoparticles (Supplementary Figure 21) synthesized by ALD (designated as Co-NPs/G-ALD) show a low selectivity (~5%) to azoxy compounds (Supplementary Figure 22)**”.

In addition, we also incorporated **Figure R1 and R2 in Supporting information as Supplementary Figure 21 and 22.**

Figure R1. TEM image of the Co-NPs/G-ALD sample. Inset shows the size distribution of Co nanoparticles. The average size of Co nanoparticles is determined to be ~4.3 nm.

Figure R2. The selectivity to azoxy compounds for different substrates over Co-NPs/G-ALD.

2. The close correlation of Co loadings with the amount of epoxy groups Figure 2g might not be able to support that the epoxy groups act as anchor sites for the Co precursors as illustrated, since the O showed in XPS might come from the oxidation of Co by ozone, rather than the formation of epoxy groups. The authors might like to explain it.

Response: We thank the reviewer for the valuable comments.

We apologize for the confusion caused and would like to clarify for this point. The XPS data presented in the supporting information was acquired over graphene support with ozone pretreatment but without the exposure of the sample to Co precursors. There is no Co atom for all the samples tested for XPS measurement. Hence, the O signal revealed by XPS measurement is attributed to the graphene support. We have included one sentence “**The graphene support is subjected to O₃ treatment without exposure to Co precursor**” in the supporting information for clarification.

3. Hydrogen dissociation in single-atom catalyst might be the rate-determining step, the authors might also like shows the ability of hydrogen dissociation on Co1 single-atoms.

Response: This is a good point. A common method for the confirmation of the hydrogen dissociation is to test the hydrogen-deuterium (H-D) exchange reaction over the catalysts. However, it is challenging to perform H-D exchange reaction to confirm the ability of hydrogen dissociation over Co₁/G SACs due to the following two reasons: (i) The hydrogen source used here is sodium borohydride (ii) the reaction is performed in the liquid phase. Nevertheless, we have conducted additional experiments to gain more information. As shown in **Table R1**, both bare graphene sample without Co loading and sodium borohydride alone are found to be inactive towards the hydrogenation of nitrobenzene. These observations further suggest that the reaction occurs over the Co site of Co₁/G SACs. It is most likely that the hydrogen radical species are generated over the Co site for this hydrogenation reaction.

Table R1. The catalytic performance of different samples for the nitrobenzene hydrogenation.

Sample	Conv. (%) ^a
Co ₁ /G SACs	99
Graphene	1
- ^b	0.5

Notes: a. Conv. represents the conversion of nitrobenzene hydrogenation; b. No catalysts.

Other minor issues:

1. On page 20, line 612, “orbitals of Co₁-C₄/G and Co₁-C₄/G.” Is there a mistake showing two times of Co₁-C₄/G?

Response: We apologize for this mistake. We have changed “orbitals of Co₁-C₄/G and Co₁-C₄/G ”into “orbitals of Co₁-C₄/G and Co₁/G”.

2. On page 8, line 259, the first appearance of “DFT+D2” in the text is not defined.

Response: This is a good point. We have defined “DFT+D2” in the revised MS.

3. In Fig. 1. The front color of “regeneration” is not very clear, it is better to change another color.

Response: We have changed the font color of “regeneration” accordingly in the revised manuscript.

4. In Fig. 5b, the structures of the two insets should be clearly defined.

Response: We have revised the two insets in the resubmitted manuscript.

Reviewer #2 (Remarks to the Author):

In this manuscript the authors employed atomic layer deposition (ALD) technique to prepare a series of Co-based single-atom catalysts (SACs) with different metal loading via controlling the number of Co ALD cycles, and investigated the catalysts for hydrogenation of nitrobenzene to azoxybenzene. The structure of CoO₂C₄ was determined with EXAFS fitting, and this Co₁/G SACs showed 68% selectivity to azoxybenzene in nitroarenes reduction using sodium borohydride (NaBH₄) as a reducing agent. DFT calculation results proposed positively charged Co₁ active centre disfavoured the adsorption of electron-deficient azoxybenzene which resulted in a higher selectivity towards azoxybenzene. My overall impression is that the paper contains rigorous material characterizations but lacks conceptual advance and novelty that are required for publication in high-level journal like Nature Communications.

Below are major concerns:

Response: Thanks for your comments. We would like to emphasize two major conceptual novelties of our work, which is also strongly supported by referee 1 and acknowledged by referee 3.

- (i) One unique finding of this study is that ozone treatment of graphene support in the second pulse of each ALD cycle not only eliminates the undesirable organic ligands of pre-deposited Co precursors, but also *recreates active sites* for the subsequent anchoring of another batch of Co atoms. Hence, the density of anchoring sites on graphene can be repeatedly generated (within the first 5 cycles tested) by controlling the number of ALD cycles, allowing for the precise tuning of the Co₁ loading of Co₁ from 0.4 wt% to 2.5 wt% without the aggregation of Co atoms. Such a high and precisely tunable loading of metal single atoms has not been achieved in the previous report. We believe that the stepwise approach developed here offers a general validity for the precise tuning of the metal loading in a wide range of single metal catalysis.
- (ii) For the first time, we also established a direct correlation between the catalytic performance of SACs and the electronic interaction between single atoms and support atoms in close proximity. Our DFT calculations reveal that the electronic coupling between Co atoms and adjacent oxygen atoms results in the partial depletion of 3d orbitals of Co atom to form a coordinately unsaturated and positively charged catalytic active centre, which disfavours the adsorption of electron-deficient azoxy compound. Such an electronic coupling between anchored Co atom and neighbouring oxygen atoms prevents the full hydrogenation of nitrobenzene, leading to a remarkable high selectivity to the partially hydrogenated product. Our findings reveal that *the proximal atoms close to metal catalytic center* play a significant role in the SACs catalysis

There have been reports on using ALD technique to prepare SACs. For example, Pt₂/graphene (ref.17), Pt₁/graphene (ref.15) and others. In fact, ALD has been proved

a powerful tool for synthesis of stable single-atom or even cluster catalysts. Since ALD has the ability to precisely control the size and distribution of particles on a substrate by using sequential and self-limiting surface reactions, it is not surprising that SACs with different loading can be obtained by precisely controlling the number of Co ALD cycles.

Response: Thanks for the reviewer's constructive suggestions.

We agree with the referee that ALD technique is a very powerful method for synthesis of ultrafine metal clusters. However, it is non-trivial to use ALD method for the synthesis of single atom catalysis and more challenging to precisely control the loading of single atoms without aggregation. To the best of our knowledge, the precise turning the loading of single atom catalysts without aggregation has not been achieved.

In the classic ALD process, the formation of *nanoparticles* and/or *clusters* are frequently observed, whereby the metal loading and size of metal particles can be tuned by controlling the ALD condition [Lu J. L., Stair P. C. *Angew. Chem. Int. Ed.* 49, 2547 (2010); Stair P. C. *Top Catal.*, 55, 43 (2012); Lu J. L., Elam J. W., Stair P. C. *Acc. Chem. Res.* 46, 1806 (2013)]. The active site on the support might be regenerated in the classic ALD process, which however, results in the formation of metal clusters or nanoparticles in most cases. Recently, the ALD has been employed for the synthesis of single atom catalysts or dimer catalysts by a dedicate control over the support as mentioned by reviewer 2. This includes Pt₂/graphene [Yan H., et al. *Nat. Commun.* 8, 1070 (2017)] and Pt₁/graphene [Cheng N., et al. *Nat. Commun.* 7, 13638 (2016)].

In general, two common models have been developed for the fabrication of SACs by ALD as shown in **Figure R3. Model 1** has been adopted to prepare the majority of SACs via ALD method, such as Pt₁ SACs [Sun S. H., et al. *Sci. Rep.* 3, 1775 (2013); Cheng N., et al. *Nat. Commun.* 7, 13638 (2016); Wang C. L., et al. *ACS Catal.* 7, 887 (2017)] and Pd₁ SACs [Yan H., et al. *J. Am. Chem. Soc.* 137, 10484(2015)];

Piernavieja-Hermida M., et al. Nanoscale 8, 15348 (2016)]. In this **Model 1**, SACs can be achieved by one **saturated** ALD cycle due to its self-limiting nature. All active sites on the support will be consumed by one **saturated** ALD cycle. In addition, **Model 2** (illustrated in **Figure R3**) has been reported in the previous work [*Yan H., et al. Nat. Commun. 8, 1070 (2017)*]. In this case, all the active sites were consumed completely in the first Pt ALD cycle and there is *no new active site* generated by the oxygen exposure. The second Pt ALD cycle deposition occurs at the anchored Pt single atoms deposited in the previous ALD cycle. This model allows the authors to achieve the synthesis of Pt dimer catalysts.

In this work, we developed a *new model* (Model 3), which is conceptually different from the previous two models. **In this new model**, the ozone treatment of graphene support in the second pulse of each ALD cycle not only eliminates the undesirable organic ligands of pre-deposited Co precursors, but also regenerates the active sites for the subsequent anchoring of another batch of Co atoms, which allows the precise tuning the density of single metal atoms without aggregation.

Figure R3. The schematic illustration of the different models for the synthesis of SACs using ALD. Notes: Model 1 and 2 have been unlisted for the synthesis of SACs in the previous work. We reported a new model in this work.

Authors proposed the atomic structure of Co centre in Co₁/G SAC was CoO₂C₄ moieties by comparing the bonding length of EXAFS fitting parameters to those in several DFT models. This method for determining the atomic structures could be ambiguous. To confirm the structure, XANES should be used and it is more sensitive to the geometrical arrangement of atoms around the centre atom (Nat. Mater., 2015, 14, 937; Nat. Commun., 2017, 8, 957; ref.30). XANES simulation of those possible DFT models to further determine the structural configuration should be more acceptable.

Response: Thanks for the reviewer's constructive suggestions.

In view of reviewer's suggestions, the calculated XANES data of various DFT modelled structures has been included into the revised manuscript. The calculated XANES curves of the proposed structures (**Figure R4a and R4c**) are in a good agreement with experimental XANES data. In contrast, the main features of experimental XANES data cannot be well reproduced in the calculated XANES data for the rest of DFT-modelled structures. Based on this additional data, we would like to point out that the proposed structures are considerably reasonable. We have revised our manuscript and placed this **Figure R4** as **Supplementary Figure 14**. In addition, we have included the **Figure R 4a and 4c** as **Figure 3b** in the main text.

Figure R4. The experimental XANES curves in comparison with the calculated XANES data of various DFT modeled structures (inset shows the atomic structures of different models).

3. The catalytic hydrogenation tests was performed by using sodium borohydride (NaBH₄) as a reducing reagent. Green reducing reagent like H₂ molecule is more interesting. And 68% selectivity to azoxybenzene was not high, many catalysts systems can reach above 90% selectivity (J.Catal., 2016, 336, 41; Green Chem., 2016, 18, 3852, and so on).

Response: Thanks for the reviewer's comments.

We find that there is scarce report on the hydrogenation to azoxy compounds (rather than azo compounds) using the green reducing reagent such as hydrogen gas.

Hence, the sodium borohydride was chosen for the hydrogenation of nitrobenzene towards azoxy compounds in our work. Indeed, there is only 68% selectivity to azoxybenzene when nitrobenzene is used as a substrate. However, Co₁/G SACs demonstrated a significantly high selectivity (90%) to azoxy aromatic compounds for other substituted nitrobenzene. Therefore, we would like to highlight that our Co₁/G SACs exhibit high selectivity in the partial hydrogenation of the majority of substituted nitrobenzene compounds.

We notice that the hydrogen sources used in the previous work mentioned by referee [*Pahalagedara M. N., et al. J.Catal. 336, 41 (2016); Zhou B., et al. Green Chem. 18, 3852 (2016)*] are hydrazine hydrate (N₂H₄·H₂O) and glucose respectively. The substrates used in one of early work is nitrosobenzene [*Zhou B., et al. Green Chem., 18, 3852 (2016)*]. A direct comparison of the selectivity in different systems may be not appropriate because the selectivity variation can arise from the utilization of different hydrogen sources and substrates. In addition, the TOF reported in these two papers is much lower than that of Co₁/G SACs reported in our work. As shown in the **Figure R5**, the TOF value of Co₁ SACs is 30 times higher than that of Ni/G nanocomposite [*Pahalagedara M. N., et al. J.Catal. 336, 41 (2016)*]. Compared with the noble metal catalysts, Co₁/G SACs also show higher TOF than that of Pd/meso CdS [*Zhou B., et al. Green Chem. 18, 3852 (2016)*]. Therefore, the Co₁/G SACs demonstrate a high activity comparable to the noble metal catalysts reported previously (**Table S2 and Figure R5**). Moreover, the TOF of Co₁/G SACs is one order of magnitude higher than that of non-precious catalysts. **We have included these two references into the Table S2 for comparison.**

Figure R5. Turnover frequency (TOF) of different catalysts.

Notes: The TOF of Pd/meso CdS and Ni/G nanocomposite is obtained from [Zhou B., et al. *Green Chem.* 18, 3852 (2016) and Pahalagedara M. N., et al. *J.Catal.* 336, 41 (2016)] respectively.

4. The substrate scope should be widely extended by using different functional and structurally diverse substrates.

Response: This is a good point. Thanks for the reviewer's comments.

In light of referee's comments, we have extended the substrate scope. As shown in the **Figure R6**, Co₁/G SACs with different loadings also exhibits a high selectivity of ~90% to azoxy compounds in the hydrogenation of other substrates, suggesting that the Co₁/G SACs show good potential for the partial hydrogenation of a broad range of substrates. **The description of catalytic performance was revised accordingly in the revised manuscript. We also included Figure R6 into the supporting information as Supplementary Figure S20. ¹HNMR and ¹³CNMR spectra of newly synthesized azoxy compounds were added into the Supporting information.**

Figure R6. The selectivity to azoxy compounds for different substrates over all the Co_1/G SACs

Reviewer #3 (Remarks to the Author):

The study deals with a new preparation method for highly dense single atom catalysts and testing these materials in a relevant reaction with important applications. I find the material synthesis and the experimental work excellent and certainly I am glad to support the work for publication. However, I have major concerns on the computational study part which I find below the state-of-the-art.

Response: We thank the reviewer's positive comments and also appreciate the reviewer's constructive suggestions.

1) The models employed are speculative and not supported by any of the experimental results. A closest match between experiment and theory is needed, otherwise the computational part can be removed without the article losing any valuable insight.

Response: We appreciate the reviewer's valuable comments.

We would like to point out that the proposed structure model for the active site ($\text{Co}_1\text{-O}_2\text{C}_4/\text{G}$) is supported by experimental data and theoretical calculations. The simulated XANES spectrum of this DFT-modelled structure shows good agreement with experimental XANES data (Figure R4c) acquired on Co_1 SACs, which offers a strong support for the proposed model. This approach has been widely used to reveal the atomic structure of active sites in SACs. In addition, our local EELS measurement provides a direct evidence to confirm the co-existence of oxygen atoms and single Co atom at the active site.

We also modelled PDOS of the active site ($\text{Co}_1\text{-O}_2\text{C}_4/\text{G}$) compared to that of a speculative structure ($\text{Co}_1\text{-C}_4/\text{G}$) for the investigation of the catalytic role of proximal oxygen atoms in the hydrogenation of nitrobenzene. We believe such a comparison helps to uncover the importance of the electronic coupling with support proximal atoms in the heterogeneous catalysis, which offers a unique opportunity to further optimize the catalytic performance of SACs.

2) Out of the different O-containing functional group only one is the epoxy with a neighboring empty cavity. As epoxides can directly exist on graphene the choice seems rather arbitrary.

Response: We appreciate the reviewer's valuable comments.

Graphene support used here has been pretreated at 300 °C for 5 min in the ALD chamber. Thermal annealing at 300 °C results in the removal of the carboxy group [Ganguly A., Sharma S., Papakonstantinou P., Hamilton J., *J. Phys. Chem. C* 115, 17009 (2011); Yan H, et al. *J. Am. Chem. Soc.* 137, 10484(2015)]]. The epoxides and/or C-OH groups are the main remaining oxygen functional groups in graphene support after thermal treatment, in agreement with the previous work [Ganguly A., Sharma S., Papakonstantinou P., Hamilton J., *J. Phys. Chem. C* 115, 17009 (2011)]]. Our XPS data (**Supplementary Figure 1**) also confirms that the ozone oxidation results in the generations of epoxides groups, in consistent with previous studies

[Mulyana Y., Uenuma M., Ishikawa Y., Uraoka Y. *J. Phys. Chem. C* 118, 27372 (2014); Gao W, et al. *Angew. Chem. Int. Ed.* 53, 3588 (2014)]. Hence, the epoxides are the predominant oxygen functional groups after thermal and the ozone pretreatment in the ALD chamber.

3) The fact that the turn-over frequency is just 6 times higher than that of nanoparticles implies that most of the Co atoms are actually non active. Can the coordination be inspected prior to use to see how much Co is trapped in the precursor form.

Response: We appreciate the reviewer's valuable comments.

If the dispersion of the Co nanoparticles is taken into account, the activity of surface Co atoms (0.30 s^{-1}) will be comparable to that of Co atoms in the Co₁/G SACs (0.33 s^{-1}) as shown in the **Table R2**. This result is also consistent with the previous work [Yan H, et al. *J. Am. Chem. Soc.* 137, 10484(2015)]. The high TOF might be mainly attributed to the atomic dispersion of Co atoms of Co₁/G SACs. In addition, the fitted XANES curve (**Figure R4**) of the calculated atomic structure of Co₁/G SACs agrees well with the experimental data, suggesting the presence of uniform and identical Co₁-O₂C₄ active sites in the Co₁/G SACs.

Table R2. The TOF based on the dispersion of catalysts.

Sample	TOF (s^{-1})
Co ₁ /G	0.33
Co-NPs/G	0.30 ^a

Notes: ^a The dispersion (d) is calculated using this equation: $d_{\text{Co}} (\%) = (1.1 / D_{\text{Co}}) \times 100$.

Note: D_{Co} refers to the size of the Co nanoparticle.

4) The active phase can be derived from computational calculations, however the paths are not presented. The work will benefit from running the whole reaction

profiles.

Response: We appreciate the reviewer's valuable comments. We agree with referee that the whole reaction profile will provide deeper insights into reaction mechanism. We did attempt to simulate the full reaction pathways. Unfortunately, the reactions are very complicated and the simulation is beyond the computer power we have. The calculated adsorption energies of azoxy compounds at different catalytic surface provide a reasonable explanation for the achieved high selectivity to azoxy products using Co₁ SACs compared to Co nanoparticles. The detailed justifications are provided in the response to the comments #5.

5) The reason for selectivity is rather speculated than inferred from the data. Either the full reaction path is calculated on the proposed structure or the claims need to be toned down.

Response: We appreciate the reviewer's valuable comments.

The hydrogenation of nitrobenzene is expected to occur in a stepwise manner including multiple steps, which involve the production of different reaction intermediates [Zhu H., et al. *Angew. Chem. Int. Ed.* 122, 9851 (2010); Hu L., et al. *Chem. Commun.* 48, 3445 (2012)] as shown in the **Figure R7**. The adsorption energy of the intermediate compounds on catalytic surface is one of the key factors that determine the selectivity to certain target products [Cheng G., et al. *Nature Mater.* 15, 564 (2016)]. In our system, we observe that azoxy compound is the major product when Co₁/G SACs is applied. This indicates that the **Step 4** (Figure R7) should be prohibited for the further hydrogenation of azoxy compound to aniline. Such a reasoning is also reported in the previous work [Xiao Q., et al. *Appl. Catal. B* 209, 69 (2017)]. We find that the adsorption energy of azoxy compounds over Co₁/G is much lower compared to both Co(111) and Co₁-C₄/G. The weak adsorption of azoxy compounds over Co₁/G might be insufficient to break the N-O bond of azoxybenzene, which is a critical step for the further hydrogenation of azoxy compounds into aniline

[Zhu H., et al. *Angew. Chem. Int. Ed.* 122, 9851 (2010); Xiao Q., et al. *Appl. Catal. B* 209, 69 (2017)]. Therefore, our calculation of adsorption energies of azoxybenzene over different catalysts provides a reasonable explanation for the high selectivity achieved in the case of Co₁/G. In the resubmitted manuscript, we have revised the claims according to the referee's suggestion.

Figure R7. The proposed reaction pathways of the hydrogenation of nitrobenzene. [Zhu H., et al. *Angew. Chem. Int. Ed.* 122, 9851 (2010); Hu L., et al. *Chem. Commun.* 48, 3445 (2012)]

6) D2 is not correct to deal with metal surfaces. The results are overbinding by a significant amount.

Reponses: thanks for your comments.

D2 has been used in many previous work for the calculation of organic molecules or carbon-based materials adsorbed on metal surfaces. We certainly agree with the referee that it is hard to control the error of D2 in these cases. The purpose of calculations presented in our work is to compare energetics of two systems (molecule adsorbed on Co(111) and Co₁/G). Hence all the calculations should be done on the

same footing. One cannot compare one adsorption energy with D2 and another one without D2. We used D2 for both situations here simply because it seems necessary to include VDW for the graphene case based on previous study [*Zhao W., et al. J. Phys. Chem. Lett.* 2, 759 (2011); *Lazar P., et al. J. Am. Chem. Soc.* 135, 6372 (2013)]

7) The description in terms of density delocalization for a van der Waals interaction is very misleading.

Response: We have revised this part in the manuscript.

8) The units for electron charge are |e-| and not e

Response: we would like to clarify that 'e' refers to electron rather than electron charge in the manuscript. To avoid this confusion, we have changed 'e' to 'electrons' in the revised manuscript.

9) The functional is misspelled.

Response: We thank the referee to point this out. We have corrected the typos.

10) 400 eV cutoff is below the VASP recommendation (415-450 eV for O)

Response: Thanks for your constructive suggestions.

This is a good point. In view of referee's comments, we have performed additional calculations for both O atom and O₂ molecules using 400 and 450 eV cut-off. Our data (**Figure R8**) show that both 400 eV and 450 eV cut-off calculations produce the same results. We also performed the calculations for the absorption energies of azoxybenzene using 450 eV cutoff in comparison with the previous results using 400 eV. Our results (**Figure R9**) reveal that there is no significant difference between two cases. Nevertheless, we have presented the results calculated using 450-eV as

suggested by referee in the revised paper.

Figure R8. The calculated DOS for O atom and O₂ molecule using 400 and 450 eV cut-off energies.

Figure R9. The calculated adsorption energies of azoxybenzene over different catalysts using different cut-off energies (400 and 450 eV).

11) The origin of the divacancy in graphene is not justified.

Response: Previous work [Yan H., et al. *Nat. Commun.* 8, 1070 (2017); Gan Y., Sun L., Banhart F., *Small* 4, 587 (2008); Rodríguez-Manzo J., Cretu O., Banhart F., *ACS Nano*. 4, 3422 (2010); Chong H. T., et al. *Science* 357, 479(2017); Deng D., *Sci. Adv.* DOI: 10.1126/sciadv.1500462] about single atom catalysts have shown that metal impurity atoms are mainly embedded in divacancies of graphene or N-doped graphene. We therefore use the divacancy in this paper for computational calculations. In addition, the calculated XANES spectra of the divacancy model agrees with our experimental XANES data very well, providing a strong justification for our proposed model. The two experimental references have been included in the revised manuscript.

12) The use of the same k-point sampling in all the calculations is wrong.

Response: thanks for your comments

The systems used for calculations contain big unit cells. We used an 8x8 unit cell for the defective graphene. For Co, a 4x4 unit cell with 5 atomic layers is used. We believe the (4x4) k-sampling should be good enough for all calculations of these structures with big unit cells. In light of referee's comments, we also performed test calculations for adsorption energy using slightly more sample points (6x6) for the Co (111) case. Compared with (4x4) k-sampling, the difference in adsorption energies is less than 1%, suggesting that the (4x4) k-sampling used in this work should be sufficient.

13) Figure 5a Ea stands for activation energy and not adsorption energy usually noted as Eads

Response: in view of referee's comments, we changed Ea to Eads in the revised manuscript.

14) The insets in Figure 5b need to be assigned in the figure.

Response: We appreciate the reviewer's valuable comments. We have assigned the insets in the Figure 5b in the new manuscript.

15) Figure 5 c-f do not represent any relevant information for the discussion they shall go to the Supplementary documentation.

Response: We appreciate the reviewer's valuable comments. We have put the Figure 5 c-f into the supporting information.

16) The description in terms of the volcano for the data presented in the study is completely irrelevant as no model for the optimum value of the adsorption is considered. The authors need to reevaluate this part.

Response: We appreciate the reviewer's valuable comments. We have rewritten this part.

Minor issues

1) The hydrogenation of nitroarenes has been reported to occur for Pt NanoSelect particles, at least a reference to this work and a comparison is needed.

Response: In view of referee's comments, we have added some references about Pt nanoparticles into the revised manuscript for comparison.

2) The authors are not considering carbon nitride scaffolds for which 1 % weight can be also obtained due to entropic contributions.

Response: We appreciate the reviewer's valuable comments. In our work, the carbon nitride is not mentioned in either experiment or calculation.

Reviewers' Comments:

Reviewer #1 (Remarks to the Author):

The authors have carefully performed additional experiments and have added substantial amount of new data into the revised manuscript. The comments raised previously have been addressed. XANES curve simulations based on the DFT-optimized structure further strengthened the catalyst structure suggested by DFT. Impressively, their Co₁ single atom catalyst also showed a high selectivity of ~90% to azoxy compounds in a broad diverse of substrates. Given that the Co₁ single atom catalyst showed significantly higher activity and selectivity even than Pt catalysts in hydrogenation of nitrobenzene aromatics with a broad diverse substitutional groups, the high loading of Co₁ singles atom with their new strategy of anchor sites regeneration during ALD synthesis, make this work very valuable to the catalysis society. With the new input, the work has been able to meet the strict requirements for publication in Nature communications, thus this reviewer recommend for acceptance.

Reviewer #2 (Remarks to the Author):

The authors made additional experimental and theoretical studies to address the concerns of the reviewers. In my opinion, all of my concerns have been addressed satisfactorily. In this case, I would like to recommend its publication in Nat Comm.

Reviewer #3 (Remarks to the Author):

The authors have addressed most of the questions I raised in previous correspondence. I have only a minor issue to be corrected.

Figure 39 the surface adsorption on Co corresponds to a metastable adsorption as the flat configuration is more stable (and cannot be correctly reproduced by the D2 results, see Scheffler Phys. Rev. Lett).

Point-by-point response letter

Reviewers' comments:

Reviewer #1 (Remarks to the Author):

The authors have carefully performed additional experiments and have added substantial amount of new data into the revised manuscript. The comments raised previously have been addressed. XANES curve simulations based on the DFT-optimized structure further strengthened the catalyst structure suggested by DFT. Impressively, their Co1 single atom catalyst also showed a high selectivity of ~90% to azoxy compounds in a broad diverse of substrates. Given that the Co1 single atom catalyst showed significantly higher activity and selectivity even than Pt catalysts in hydrogenation of nitrobenzene aromatics with a broad diverse substitutional groups, the high loading of Co1 singles atom with their new strategy of anchor sites regeneration during ALD synthesis, make this work very valuable to the catalysis society. With the new input, the work has been able to meet the strict requirements for publication in Nature communications, thus this reviewer recommend for acceptance.

Response: We highly appreciate reviewer 1 for the valuable comments and support for the publication of our work.

Reviewer #2 (Remarks to the Author):

The authors made additional experimental and theoretical studies to address the concerns of the reviewers. In my opinion, all of my concerns have been addressed satisfactorily. In this case, I would like to recommend its publication in Nat Comm.

Response: We would like to thank reviewer 2 for the valuable comments and recommendation of the publication of our manuscript.

Reviewer #3 (Remarks to the Author):

The authors have addressed most of the questions I raised in previous correspondence. I have only a minor issue to be corrected. Figure 39 the surface adsorption on Co

corresponds to a metastable adsorption as the flat configuration is more stable (and cannot be correctly reproduced by the D2 results, see Scheffler Phys. Rev. Lett).

Response: we thank reviewer 3 for this valuable comment.

We would like to point out that the aim of our calculation is to compare the adsorption energy of the azoxybenzene molecule on Co (111) surface and on Co₁/G support. We noted that the combination of DFT + vdW and the LKZ theory [Ruiz VG, Liu W, Zojer E, Scheffler M, Tkatchenko A. Phys. Rev. Lett. 108, 146103 (2012)] provides a more accurate prediction of the adsorption configurations of organic molecules on metal surfaces. While we do not disagree with the referee that the adsorption configuration of azoxybenzene on Co predicted by DFT + D2 (presented in Figure 39) could be a meta-stable structure, we also found that the adsorption energy of this structure on Co(111) (-1.15 eV) is already much more negative compared to that on Co₁/G (-0.52 eV). Hence, the conclusion that azoxybenzene shows a weaker adsorption on the Co₁/G support compared to Co (111) is still valid. In addition, previous studies have revealed that it is necessary to include vdW in the DFT calculations of the graphene system. The resulting calculation results show good agreement with experimental data [Zhao W., et al. J. Phys. Chem. Lett. 2, 759 (2011)]; [Lazar P., et al. J. Am. Chem. Soc. 135, 6372 (2013)]. In order to have a direct comparison of the adsorption energies of azoxybenzene on both substrates, we therefore performed DFT+vdW (in D₂ format) calculations for the above-mentioned two cases.